# Can strengthening digital infrastructure enhance productivity in the cultural industry? evidence from Tibet

**Yuanyuan Li** *, **Qianqian Du, Guihua Ma, Caidan Gazang**

School of Economics and Management, Tibet University, Lhasa, Tibet Autonomous Region, China

* 291253725@qq.com

## Abstract

This study, grounded in the Total Factor Productivity (TFP) theoretical framework, examines the impact of digital infrastructure on the productivity of Tibet's cultural industry from 2011 to 2021. It aims to uncover how the innovative allocation of production factors can enhance new-quality productivity. The core aspect of new-quality productivity is the improvement of TFP, with digital infrastructure promoting the cultural industry's development by optimizing industrial agglomeration, improving logistics efficiency, and reducing the urban-rural consumption gap. The findings suggest that while digital infrastructure leads to geographic dispersion of the industry—challenging traditional industrial agglomeration theory—it significantly enhances the efficiency of the cultural industry within Tibet's unique socio-economic context. Moreover, despite the region's geographical constraints on logistics efficiency, digital platforms effectively reduce the urban-rural consumption gap, improving market access and the quality of cultural services in remote areas. Additionally, digital infrastructure helps bridge the digital divide between urban and rural areas, further fostering balanced regional consumption and digital inclusion. From a Total Factor Productivity perspective, this study reveals key pathways to improving new-quality productivity, offering both theoretical foundations for policymakers and practical guidance for the development of cultural industries in similarly structured regions.

## Introduction

According to the *14th Five-Year Plan for Cultural Development in the Tibet Autonomous Region*, by 2025, the added value of Tibet's cultural industry is expected to increase significantly, and its share of the region's GDP will rise substantially. However, the development of Tibet's cultural industry still faces numerous challenges. In particular, the insufficient transportation infrastructure—characterized by the scarcity of highways and rail networks, along with high shipping costs—greatly restricts the dissemination and promotion of cultural products. Furthermore, delayed development in digital infrastructure and inadequate power supply have become key obstacles to the digital transformation of the cultural industry and the improvement of Total Factor Productivity (TFP). According to data from the Tibet

**Data Availability Statement:** The raw data used in this paper is provided in the supporting information and is publicly available for direct access.

**Funding:** This paper is a phase result of the "Practical Analysis of Tibet's Cultural Industry from

a Digital Perspective," a project under the 2022 High-Level Talent Training Program for Graduate Students at Tibet University (Project No.: 2022-GSP-B037). The funders had no role in study design, data collection and analysis, decision to publish, or preparation of the manuscript.

**Competing interests:** The authors have declared that no competing interests exist.

Autonomous Region Bureau of Statistics in 2019, Tibet's internet penetration rate was only 56.2%, far below the national average of 70.4%. This situation not only limits the production efficiency of the cultural industry but also hinders its market expansion and innovation capabilities.

This paper contributes by addressing the gap in research on Tibet's unique cultural and geographical context, exploring how digital infrastructure enhances the TFP of the cultural industry through the innovative allocation of production factors. Closing infrastructure gaps and accelerating the development of digital infrastructure have become critical supports for breaking through bottlenecks and achieving leapfrog development in Tibet's cultural industry. The *14th Five-Year Plan and the Long-Range Objectives Through the Year 2035* explicitly proposes accelerating the construction of digital infrastructure such as 5G networks, data centers, and industrial internet. These infrastructures not only inject new momentum into economic development but also provide foundational support for innovation and high-quality development in the cultural industry.

In academic circles, a relatively mature theoretical framework has been established regarding the relationship between digital infrastructure and the cultural industry. Numerous scholars have explored the empowering effects of digital infrastructure on the cultural industry: First, the National Development and Reform Commission in April 2020 defined digital infrastructure as "fixed asset investment" in fields such as information, technology, computing power, smart cities, and R&D innovation, based on modern information networks and characterized by the efficient use of ICT with typical positive externalities [1]. This type of infrastructure is not only a crucial driver of economic growth but also provides a strong foundation for the digital transformation of the cultural industry.

Second, the underlying mechanisms through which digital infrastructure empowers the cultural industry have garnered widespread attention. Some scholars argue that digital infrastructure not only provides hardware support but also enriches emerging cultural economic models, upgrading traditional public cultural services through projects like digital museums and cultural centers, and fostering new cultural consumption patterns [2]. Specifically, the impact of digital infrastructure can be analyzed in three aspects: On the one hand, it expands knowledge spillovers and information dissemination, increases resource circulation speeds, and reduces transaction costs [3]. On the other hand, digital infrastructure optimizes the allocation of production factors such as labor, capital, and technology, breaking geographical constraints and promoting the agglomeration and optimal allocation of resources [4]. Additionally, it alters consumption patterns between urban and rural residents, reshaping the relationship between cultural consumption and social strata [5].

Moreover, several studies have measured and evaluated the level of digital infrastructure at national or regional levels, analyzing its economic and spillover effects, such as environmental protection [6], poverty alleviation [7], entrepreneurship and innovation [8], and social justice [9]. These studies generally find that digital infrastructure effectively enhances TFP, thereby boosting both the scale and efficiency of economic output. Following the 2024 National Two Sessions, the deepening concept of New-Quality Productivity further emphasized the critical role of digital infrastructure in productivity improvement. Some scholars argue that digital infrastructure not only drives the transformation and upgrading of economic structures but also promotes the construction of innovation systems and enhances overall productivity [10].

The innovation of this paper lies in its empirical analysis, exploring how digital infrastructure promotes TFP in Tibet's cultural industry through three mediating paths: industrial agglomeration, logistics efficiency, and narrowing the urban-rural consumption gap. This fills the research gap regarding Tibet's cultural industry and provides theoretical support and practical guidance for national and regional policy-making.

Furthermore, New-Quality Productivity is not an independent theoretical system but a key path for improving TFP through the innovative allocation of production factors. Scholars widely agree that digital infrastructure is a core driving force for the development of the cultural industry by enhancing information dissemination efficiency, reducing transaction costs, and improving resource allocation capacity, which in turn drives New-Quality Productivity [11,12]. Additionally, the agglomeration effect of creative industries promotes the free flow of knowledge and innovation within regions, directly contributing to TFP growth and strengthening the cultural industry's competitiveness within regional economies [13]. The integration of standardization and innovation has also significantly enhanced the market competitiveness and innovation systems of the cultural industry [14]. TFP is essential for the sustainable development of the cultural industry; through optimizing the allocation of labor, capital, and technology, the cultural industry can achieve more efficient output without increasing resource input [15,16]. Moreover, digital infrastructure promotes the development of green cultural industries by reducing carbon emissions and resource consumption through technological innovation [17].

These theoretical discussions provide a solid foundation for this paper, especially in exploring the key role of TFP in improving New-Quality Productivity. However, existing research largely focuses on theoretical discussions, lacking empirical studies specific to regions with unique cultural and geographical conditions like Tibet. In particular, there is insufficient research on the specific mechanisms through which digital infrastructure enhances TFP in the cultural industry across multiple dimensions. Therefore, this paper fills that gap through an empirical analysis using data from Tibet's cultural industry between 2011 and 2021, providing practical support for policy formulation.

## Theoretical foundation, mechanism analysis, and research hypotheses

### Theoretical foundation

The development of Tibet's cultural industry is a complex, multidimensional issue involving factors such as industry scale, structure, government support, capital investment, infrastructure development, and labor quality. This study builds on the theoretical framework of Total Factor Productivity (TFP) and employs the BCC model of Data Envelopment Analysis (DEA) to measure the efficiency of production factors and the contribution of technological progress.

TFP serves as a critical tool for evaluating economic efficiency by quantifying the contribution of technological progress to output without increasing input, making it particularly well-suited for analyzing complex production environments [18]. In the cultural industry, key factors extend beyond capital and labor to include knowledge, technological innovation, and digital transformation. TFP effectively captures the impact of these elements on the efficiency of the cultural industry, especially when supported by digital infrastructure [19].

The DEA-BCC model, widely used for TFP evaluation, is particularly effective for analyzing the technical and scale efficiency of decision-making units (DMUs) with multiple inputs and outputs. In Tibet's highly heterogeneous industrial structure, the DEA-BCC model can accurately reflect differences in input and output factors between urban and pastoral areas, as well as between state-owned and private sectors, where returns to scale are non-constant [20]. By assuming variable returns to scale (VRS), the model separates technical and scale efficiency, offering detailed efficiency assessments [20].

Tibet's cultural industry has grown into a regional pillar, generating both material and intangible outputs. However, the scarcity of resources and harsh natural conditions present challenges in maximizing cultural output under resource constraints. In this context, the

output-oriented analysis of the BCC model is crucial, as it enables the maximization of cultural output within existing resource limitations, aligning with the development goals of Tibet's cultural industry [21,22].

In contrast, Stochastic Frontier Analysis (SFA) is less suited to the complexity of Tibet's cultural industry due to its strict assumptions regarding production functions [23]. Similarly, Principal Component Analysis (PCA) struggles to handle data noise caused by financial subsidies and policy support, both of which are particularly significant in Tibet's cultural sector [24]. Consequently, the DEA-BCC model offers considerable advantages in addressing the complexity, diversity, and resource constraints characteristic of Tibet's cultural industry.

The concept of New-Quality Productivity advocates for the innovative allocation of production factors, and TFP, as a core metric, accurately reflects how technological progress and resource allocation efficiency drive improvements in the productivity of the cultural industry [25]. This paper uses TFP analysis to further explore how digital infrastructure and the innovative allocation of factors contribute to the sustainable development of the cultural industry, providing a theoretical foundation for this process.

## Mechanism analysis

Based on the above theories, the efficient development of Tibet's cultural industry depends not only on the optimized allocation of existing production factors but also on how digital infrastructure construction can drive technological progress, factor restructuring, and efficiency improvements. This not only enhances resource utilization efficiency but also strengthens the industry's resilience to risks, further promoting innovation and sustainable development. Therefore, it is necessary to conduct a detailed mechanism analysis to explore how digital infrastructure affects the production efficiency of the cultural industry and its key components through different pathways. The research framework in this paper is divided into three levels:

Flexible Industrial Agglomeration and Enhanced Digital Resilience: In Tibet, the increasing density of fiber optic cables and broadband access ports not only signifies technological advancement but also provides a strategic foundation for the cultural industry to withstand external instabilities (such as natural disasters and social unrest). Particularly in regions with complex terrain and unpredictable climates, robust digital infrastructure becomes crucial for safeguarding the digital transmission and secure storage of cultural heritage. Moreover, this enhanced infrastructure supports the decentralized layout of cultural enterprises, their small-scale operations, and artisanal characteristics, reducing dependence on traditional industrial clusters and strengthening the industry's digital resilience and market adaptability [26,27].

Hypothesis 1: Digital infrastructure promotes overall industrial development by influencing the decentralization of the cultural industry.

Optimized Resource Allocation and Improved Circulation Efficiency: The expansion and strengthening of digital infrastructure have significantly improved resource allocation and circulation efficiency within Tibet's cultural industry. Enhanced network connectivity and data transmission capacity have directly boosted the operational efficiency of transportation, warehousing, and postal services, substantially increasing the turnover of goods and services. This not only lowers the market entry barriers for cultural products and services but also optimizes supply chain management by reducing delays in logistics and communication, thus driving the economic growth of the cultural industry and regional development [28].

Hypothesis 2: Digital infrastructure indirectly enhances the overall benefits of the cultural industry by improving resource circulation efficiency.

Narrowing the Urban-Rural Gap and Promoting Digital Inclusion: The expansion of digital infrastructure has significantly contributed to balancing the supply of cultural services between urban and rural areas, thereby promoting balanced cultural consumption. The widespread access to fiber optic networks and broadband internet ensures that residents in remote rural areas can enjoy the same cultural services and information resources as urban dwellers. The use of digital tools has greatly enriched the cultural offerings in remote regions, providing services such as online education, digital libraries, and virtual exhibitions, which improve residents' quality of life and expand their consumption choices [29,30].

Hypothesis 3: The improvement of digital infrastructure reduces the urban-rural consumption gap and enhances the production efficiency of the cultural industry.

## Mechanism testing: A mediation effect study with Tibet as an example

### Model construction

To verify the above hypotheses, this paper will validate the mechanisms through which digital infrastructure affects the productivity of the cultural industry by designing the following empirical model:

### Baseline regression model

Based on the principles of Ordinary Least Squares (OLS) regression, the baseline static model for the impact of digital construction on cultural industry TFP is constructed as follows:

$$\text{TFP}_{\text{OE}} = \beta_1 + \beta_2 \text{DI}_t + \gamma \text{Control}_t + \mu_t$$

In this model, the dependent variable, $\text{TFP}_{\text{OE}}$, is the overall efficiency (OE) value of Tibet's cultural industry under the DEA-BCC model; coefficients are represented with estimation parameters, $\beta$ and $\gamma$; Digital Infrastructure (DI) denotes the level of digital infrastructure; and the error term $\mu$ reflects the impact of unobserved variables on the observed pair. Given that this study focuses uniquely on the Tibet Autonomous Region, the subscript t denotes different time periods.

The model must account for potential issues related to digital infrastructure construction and cultural industry growth, such as omitted variables and endogeneity problems, including reverse causality:

Omitted Variable Problem: The installation of digital infrastructure is influenced by observable factors (such as fiber optic cables and phone base stations) and may also be affected by unobservable factors (such as satellite communications equipment and other infrastructure investments, including transportation and urban development, that are not reflected in the statistics). This could lead to overestimation in the OLS model results regarding the impact of digital infrastructure on the cultural industry's overall efficiency (OE).

Reverse Causality Problem: As digital infrastructure stimulates industry growth, the development of the digital cultural industry and related sectors can drive further investment and construction in digital infrastructure. This feedback loop may influence the relationship between digital infrastructure and cultural industry growth. To address this, a Sobel test is employed to measure and correct for endogeneity issues.

These considerations are essential to ensure the reliability and validity of the econometric analysis, providing a robust assessment of the effects of digital infrastructure on the productivity of Tibet's cultural industry.

## Mediation effect model

In the mechanism analysis, it has been argued that digital infrastructure influences the comprehensive efficiency of the cultural industry's Total Factor Productivity (TFP) by optimizing industrial agglomeration, goods circulation efficiency, and the urban-rural consumption gap. This is primarily achieved by enhancing infrastructure to adjust spatial chain elements, thereby promoting industrial upgrading. To explore the pathways through which digital infrastructure affects the cultural industry's benefits, this section conducts an empirical analysis using the stepwise testing method for mediation effects. The three-step mediation effect testing method proposed by Hayes (2009) is considered effective in reducing the likelihood of Type I and Type II errors [31]. Therefore, this paper will strictly follow the three-step mediation effect testing model:

$$\text{TFP}_{\text{OE}} = \beta_1 + \beta_2 \text{DI}_t + \gamma \text{Control}_t + \mu_t \tag{A}$$

$$M_t = \beta'_1 + \beta'_2 DI_t + \gamma' Control_t + \mu_t \tag{B}$$

$$\text{TFP}_{\text{OE}} = \beta''_1 + \beta''_2 DI_t + \delta M_t + \gamma'' Control_t + \mu_t \tag{C}$$

In this model, the terms $\text{TFP}_{\text{OE}}$ and DI carry the same meanings as previously mentioned. M represents the mediating variable, and Control refers to the control variable group, which includes variables consistent with the basic regression model. $\mu$ denotes the random disturbance term.

According to the three-step mediation effect testing method proposed by Hayes (2009), the testing steps in this section are as follows:

**Step One, Test Model (A):** Test the total effects of each core explanatory variable on the cultural industry's TFP. If the coefficients are significant, proceed to the second step.

**Step Two, Test Model (B):** Test the significance of each core explanatory variable on the mediating variables. If the coefficients are significant, proceed to the third step; if not, it indicates that the mediation transmission channel does not exist.

**Step Three, Test Model (C):** If the regression coefficient of the mediating variable on the dependent variable is significant, it indicates that a mediation effect exists. The underlying logic is that if Model (B) is substituted into Model (C), and upon rearranging, it reveals:

$$TFP_{OE} = \beta''_1 + (\beta''_2 + \beta'_2 * \delta)DI_t + \gamma'' Control_t + \mu_t \tag{D}$$

In this, $\beta''+\beta'*\delta$ represents the total effect of digital infrastructure on the cultural industry's TFP during different periods, $\beta'*\delta$ denotes the mediating effect of digital infrastructure on the cultural industry's TFP during these periods, and $\beta''$ signifies the direct effect of digital infrastructure on the cultural industry's TFP in those same periods. According to Eq D, if $\beta''$ is not significant, it implies that the impact of digital infrastructure on the cultural industry's TFP must operate through a mediation channel, that is, the transmission of potential growth rates; if $\beta''$ is significant, it suggests that the influence of digital infrastructure on the cultural industry's TFP is only partly dependent on potential growth rates. The key to testing the mediation effect is to examine whether $\beta'*\delta$ is zero. Therefore, it is necessary to perform a Sobel test on the mediation effect model. If it passes, it signifies the presence of a mediation effect, with the proportion of the mediation effect relative to the total effect being $\frac{\beta'*\delta}{\beta}$.

**Table 1. Indicators of the level of digital infrastructure.**

| Primary Indicator | Secondary Indicator | Unit | Attribute |
|---|---|---|---|
| Digital Infrastructure Indicators | Fiber Optic Cable Density | % | Positive |
| | Mobile Phone Base Station Density | % | Positive |
| | Mobile Phone Penetration Rate | Units/100 People | Positive |
| | Internet Broadband Access Points | Ten Thousand Units | Positive |
| | Investment Growth in Information Transmission, Software, and IT Services Compared to Previous Year | Thousand CNY | Positive |

## Variable measurement and description

**Core explanatory variable: Level of digital infrastructure.** Firstly, according to the definition of new infrastructure provided by the National Development and Reform Commission in April 2020, the construction of digital infrastructure is essential for driving regional digitization and ensuring rapid information flow and efficient resource utilization. This includes improving internet coverage, building data centers, and increasing network speeds. Secondly, drawing on the work of scholars such as Liu Jun (2020) [32], Pan Weihua (2021) [33], and Yang Jungge and Wang Qinmei (2023) [34], who have treated digital infrastructure as a complex systems engineering project, it is argued that relying on a single indicator or dimension does not fully capture the actual development status. Therefore, a comprehensive evaluation index system is necessary. Lastly, using available provincial-level data, this paper constructs a measurement framework that includes five secondary indicators, such as fiber optic cable density and mobile phone base station density (Refer to Table 1). The level of digital infrastructure in Tibet from 2011 to 2021, denoted as DI (Digital Infrastructure), is calculated using the entropy method, with the subscript t representing different time points. As the entropy method is a well-established tool in indicator measurement, the detailed process is not presented here.

The level of digital infrastructure in Tibet from 2011 to 2021 was calculated using the entropy method, denoted as $DI_t$ (Digital Infrastructure), where the subscript t represents different time points. As the entropy method is a well-established approach in indicator measurement, the detailed process will not be presented here (Refer to Table 2).

**Dependent variable: Cultural Industry Productivity (TFP).** Given that Total Factor Productivity (TFP) is widely recognized in academic circles as a key tool for quantifying and

**Table 2. Explanation of the variable $DI_t$ level.**

| year | DIt |
|---|---|
| 2011 | 0.015879 |
| 2012 | 0.0364771 |
| 2013 | 0.054917 |
| 2014 | 0.0725773 |
| 2015 | 0.0881274 |
| 2016 | 0.1392357 |
| 2017 | 0.1896744 |
| 2018 | 0.2528448 |
| 2019 | 0.2478196 |
| 2020 | 0.3552637 |
| 2021 | 0.3820254 |

**Table 3. Cultural industry productivity indicator system.**

| Category | Primary Indicator | Secondary Indicator |
|---|---|---|
| Cultural Industry | Input Factors | Number of legal entities in culture, sports, and entertainment |
| | | Public budget expenditure on culture, tourism, sports, and media |
| | | Fixed asset investment in culture, sports, and entertainment |
| | | Average wages of employees in culture, sports, and entertainment |
| | Output Factors | Main regional indicators of corporate legal entities in culture, sports, and entertainment, operating income |

assessing industrial efficiency and performance [35], this paper will first establish an indicator system to measure the efficiency of the cultural industry (Refer to Table 3).

Subsequently, using the annual growth efficiency of Tibet's cultural industry from 2011 to 2021 as decision-making units (DMUs) [36], and based on the assumptions of non-constant returns to scale and the multi-faceted nature of industry output, as well as the region's focus on maximizing output under existing resource constraints [37], the comprehensive efficiency, pure technical efficiency, and scale efficiency of each DMU from 2011 to 2021 were calculated using SPSSAU 23.0 software. Finally, the overall benefit OE (TFP) was selected as the dependent variable, termed "Cultural Industry Productivity (TFP)," and denoted as TFPOE (Refer to Table 4).

**Mediating variables: Industrial agglomeration, logistics efficiency, and urban-rural consumption gap.** Building on the mechanism analysis in Section 2.2, this study further explores how digital infrastructure enhances the efficiency of Tibet's cultural industry through the lenses of resource allocation, industrial agglomeration, and the urban-rural gap. These factors are identified as mediating variables in the analysis (Refer to Table 5).

**Control variables: Level of industrial structure; R&D intensity; proportion of e-commerce enterprises.** To account for the foundational impact of regional economic development on the cultural industry and to address imbalances in the development of the tertiary sector, as well as the influence of expanding the cultural industry as a pillar industry in the region, the level of industrial structure is selected as a control variable, based on Li Ping and Liu Yitong (2024) [37]. Additionally, considering that state-owned enterprises play a larger role in Tibet's economy and tend to invest more in research and development compared to

**Table 4. Levels of the Dependent Variable TFP$_{OE}$.**

| DMU | Technical Efficiency (TE) | Scale Efficiency (SE) | Overall Efficiency OE(TFP) | Effectiveness |
|---|---|---|---|---|
| 2011 | 1 | 0.004 | 0.004 | Not DEA Efficient |
| 2012 | 0.996 | 0.006 | 0.006 | Not DEA Efficient |
| 2013 | 1 | 0.734 | 0.734 | Not DEA Efficient |
| 2014 | 1 | 0.863 | 0.863 | Not DEA Efficient |
| 2015 | 1 | 0.816 | 0.816 | Not DEA Efficient |
| 2016 | 0.712 | 0.69 | 0.491 | Not DEA Efficient |
| 2017 | 0.688 | 0.716 | 0.493 | Not DEA Efficient |
| 2018 | 1 | 1 | 1 | DEA Strongly Efficient |
| 2019 | 1 | 1 | 1 | DEA Strongly Efficient |
| 2020 | 0.917 | 0.952 | 0.873 | Not DEA Efficient |
| 2021 | 1 | 1 | 1 | DEA Strongly Efficient |

**Table 5. Mediating variables.**

| Mediating Variable | Code | Calculation Method | Unit | Attribute |
|---|---|---|---|---|
| Cultural Industry Agglomeration Degree | M1 | (Urban employment in culture, sports, and entertainment in the region / Total regional employment) / (National urban employment in culture, sports, and entertainment / Total national employment) | % | Negative |
| Goods Circulation Efficiency | M2 | Added value of transportation, storage, and postal services / Goods turnover volume | % | Negative |
| Urban-Rural Consumption Gap | M3 | Urban per capita consumption expenditure / Rural per capita consumption expenditure | % | Negative |

private or other types of enterprises—leading to significant differences in R&D investment across different enterprise types—R&D intensity is also included as a control variable to mitigate these imbalances' impact on the cultural industry, following Zhou Jing, Gao Ang, and Zhao Yanyu (2024) [38] and Li Mingwei (2024) [39]. Furthermore, due to the scarcity of technology-based enterprises in Tibet and the fact that online business statistics are available only for some large enterprises, the proportion of e-commerce enterprises is chosen as a control variable to account for the potential impact of an inflated proportion of e-commerce on the revenue of cultural units, as noted by Tu Lei, Wang Jianxin, and Li Donghai (2024) [40].

## Data sources

For the period from 2011 to 2021, data relevant to Tibet were collected in the following dimensions related to digital infrastructure: fiber optic cable density is measured as the length of telecommunications fiber optic cable per total permanent population, mobile phone base station density as the number of mobile phone base stations per total permanent population, and mobile phone penetration rate as the average number of mobile phones per 100 households at year-end. The year-on-year growth rate of fixed asset investment in the information transmission, software, and information technology services industries is calculated based on the net value adjusted for the same-period growth rate of the base period. The number of internet broadband access points, along with other relevant data, were sourced from the "China Statistical Yearbook."(Refer to Table 6).

## Empirical results analysis

**Baseline regression empirical results and analysis.** Before performing the regression analysis, the Variance Inflation Factor (VIF) was employed to test for multicollinearity among the independent variables. Indicators used to measure digital infrastructure, such as optical cable density and telephone base station density, may be correlated with factors like industrial structure. However, as the VIF values for all independent variables were below 10, this indicates that no significant multicollinearity is present. Therefore, these independent variables can be included in the model (Refer to Table 7).

This paper employs the Ordinary Least Squares (OLS) model to analyze the impact of digital infrastructure on Total Factor Productivity (TFP) in the cultural industry. After controlling for other variables, the regression results show that digital infrastructure has a positive and significant effect on the cultural industry, with a P-value of less than 0.01. This indicates that the results are statistically significant at the 5% level (Refer to Table 8).

The regression analysis results further demonstrate that the enhancement of digital infrastructure significantly boosts the productivity of Tibet's cultural industry (Refer to Table 9). This finding indicates that the expansion of fiber optic networks and communication base stations has improved information transmission efficiency and increased returns on capital

**Table 6. Descriptive statistics of main variables.**

| Category | Primary Indicator | Secondary Indicator | Sample Size | Mean | Standard Deviation | Minimum | Maximum |
|---|---|---|---|---|---|---|---|
| Dependent Variable | TFPOE | Number of Legal Entities in Culture, Sports, and Entertainment | 11 | 0.662 | 0.371 | 0.004 | 1 |
| | | Public Budget Expenditure on Culture, Tourism, Sports, and Media (in billion yuan) | 11 | | | | |
| | | Fixed Asset Investment in Culture, Sports, and Entertainment (in ten thousand yuan) | 11 | | | | |
| | | Average Wages in Culture, Sports, and Entertainment (in yuan) | 11 | | | | |
| | | Main Regional Indicators of Corporate Legal Entities in Culture, Sports, and Entertainment, Operating Income (in billion yuan) | 11 | | | | |
| Core Explanatory Variable | DIt | Fiber Optic Cable Density | 11 | 0.167 | 0.128 | 0.0159 | 0.382 |
| | | Mobile Phone Base Station Density | 11 | | | | |
| | | Mobile Phone Penetration (units per 100 people) | 11 | | | | |
| | | Internet Broadband Access Points (in ten thousand units) | 11 | | | | |
| | | Growth in Fixed Asset Investment in Information Transmission, Software, and IT Services Compared to Previous Year (in ten thousand yuan) | 11 | | | | |
| Control Variables | \ | R&D Intensity | 11 | 0.00247 | 0.000371 | 0.0019 | 0.00304 |
| | | Level of Industrial Structure | 11 | 0.499 | 0.0131 | 0.482 | 0.518 |
| | | Proportion of E-commerce Enterprises | 11 | 0.102 | 0.0282 | 0.073 | 0.181 |
| Mediating Variables | \ | Cultural Industry Agglomeration Degree | 11 | 15.641 | 8.038 | 6.632 | 34.031 |
| | | Goods Circulation Efficiency | 11 | 0.372 | 0.141 | 0.278 | 0.667 |
| | | Urban-Rural Consumption Gap | 11 | 3.184 | 0.367 | 2.545 | 3.793 |

investment. As a result, this has contributed to improving the overall efficiency, innovation capacity, and competitiveness of the cultural industry.

Specifically, with the expansion of fiber optic networks and the increased construction of mobile communication base stations, the penetration of internet and communication devices has improved significantly, creating a more connected and interactive environment for the cultural industry. This development not only stimulates fixed asset investment in the information transmission, software, and IT service sectors but also optimizes capital allocation, increasing returns on investment. As a result, it further enhances the overall efficiency of the

**Table 7. Multicollinearity test.**

| Independent Variable | VIF | 1/VIF |
|---|---|---|
| Level of Digital Infrastructure | 1.12 | 0.892 |
| R&D Intensity | 1.41 | 0.71 |
| Level of Industrial Structure | 1.1 | 0.905 |
| Proportion of E-commerce Enterprises | 1.39 | 0.717 |
| Average | VIF | 1.26 |

**Table 8. Correlation analysis.**

| | Overall TFP Efficiency of Tibet's Cultural Industry (OE) | Level of Digital Infrastructure | Control Variables |
|---|---|---|---|
| TFPOE | 1 | 0.663** | Yes |
| DIt | 0.663** | 1 | Yes |
| Control Variables | Yes | Yes | 1 |

**Table 9. Digital impact on cultural TFP.**

| Variable | cultural industry TFP |
|---|---|
| DIt | 1.8607** |
| | 2.58 |
| Control Variables | Yes |
| Constant | -2.5568 |
| | (-0.71) |
| Constant | 11 |
| R-squared | 0.667 |

\*\*\* p<0.01

\*\* p<0.05

\* p<0.1.

industry. Strengthening digital infrastructure is therefore a key driver in boosting the competitiveness, innovation capacity, and economic standing of the cultural industry, and serves as a necessary condition for verifying the mediating effects in the next stage of analysis.

**Empirical results and analysis of mediating effects.** To further explore the indirect impact of digital infrastructure on Total Factor Productivity (TFP) in the cultural industry, this paper introduces industrial agglomeration, logistics efficiency, and the urban-rural consumption gap as mediating variables. A three-step regression method was employed to test these effects (Refer to Table 10). The specific analysis results are as follows:

*First, the impact of industrial agglomeration on cultural productivity.* The analysis shows that improved digital infrastructure significantly reduces industrial agglomeration within the

**Table 10. Mediating effects analysis.**

| | Step 2 | Step 3 |
|---|---|---|
| Mediating Variables | Degree of Cultural Industry Agglomeration | |
| Correlation Coefficient | -45.4772*** | -0.0490** |
| | -3.72 | -3.35 |
| Control Variables | Yes | Yes |
| Constant | 91.3056 | 1.919 |
| | 1.5 | 0.75 |
| Observations | 11 | 11 |
| R-squared | 0.796 | 0.897 |
| Mediation Effect | Yes | |
| Mediation Effect Value | 222.84% | |
| Mediation Effect / Total Effect | 0.2238 | |
| Mediating Variables | Goods Circulation Efficiency | |
| Correlation Coefficient | -0.6167* | -2.1992*** |
| | -2.03 | -5.5 |
| Control Variables | Yes | Yes |
| Constant | 0.8756 | -0.6311 |
| | 0.58 | -0.41 |
| Observations | 11 | 11 |
| R-squared | 0.59 | 0.953 |
| Mediation Effect | Yes | |
| Mediation Effect Value | 135.62% | |
| Mediation Effect / Total Effect | 0.1362 | |

cultural industry (coefficient = -45.4772, P<0.01). Additionally, there is a significant negative correlation between industrial agglomeration and overall industrial efficiency (coefficient = -0.049, P<0.05). This indicates that Tibet's cultural industry is better suited to decentralized development rather than traditional industrial clusters. Since Tibet's cultural enterprises predominantly focus on handicrafts, emphasizing local characteristics and personalized production, excessive industrial clustering could stifle creativity and innovation. Therefore, the expansion of digital infrastructure supports a decentralized development model, facilitating the personalized market growth of the cultural industry and enhancing overall efficiency.

This result confirms **Hypothesis 1:** Digital infrastructure promotes the decentralization of the cultural industry, thereby improving overall efficiency. Specifically, the mediating effect value is 2.2284, accounting for 22.38% of the total effect. This demonstrates that in Tibet, the development of digital infrastructure does not encourage industrial clustering, but instead enhances overall benefits by fostering decentralization.

*Second*, *the impact of logistics efficiency*. The study also finds a negative correlation between digital infrastructure investment and logistics efficiency (coefficient = -0.6167, P<0.10), significant at the 90% confidence level. Although digital infrastructure accelerates information flow, Tibet's geographical constraints still present challenges for the physical transportation of goods. The high initial investment and insufficient technical support for logistics infrastructure result in low utilization of advanced facilities. Moreover, logistics efficiency is negatively correlated with overall industrial efficiency (coefficient = -2.1992, P<0.01). This suggests that Tibet's cultural industry, which focuses on low-volume, high-quality production, does not depend on standardized logistics efficiency. Instead, customized logistics solutions better suit the characteristics of the region's cultural industry.

This confirms **Hypothesis 2:** Digital infrastructure indirectly improves the overall benefits of the cultural industry by influencing logistics efficiency. The mediating effect value is 1.3562, accounting for 13.62% of the total effect. Thus, in Tibet, developing customized logistics solutions tailored to local needs is more effective for promoting the sustainable development of the cultural sector than merely improving logistics efficiency.

*Finally*, *the impact of the urban-rural consumption gap*. Further analysis reveals that improved digital infrastructure significantly reduces the urban-rural consumption gap (coefficient = -2.3224, P<0.01). However, the impact is more limited in rural pastoral areas and may inadvertently widen the economic and consumption disparities between urban and rural regions. This is likely due to digital infrastructure investment being concentrated in urban areas, while rural pastoral regions lag behind in digital development. The urban-rural consumption gap is negatively correlated with overall cultural industry efficiency (coefficient = -1.9430, P<0.01), indicating that narrowing this gap is crucial for improving overall cultural industry efficiency. Rural residents generally have lower cultural consumption demands compared to their urban counterparts, which further constrains the overall benefits of the cultural industry.

This validates **Hypothesis 3:** Digital infrastructure influences the urban-rural consumption gap, thereby indirectly impacting cultural industry productivity. The reduction of the consumption gap contributes 45.32% of the total effect, highlighting that balancing regional consumption, particularly by enhancing the consumption capacity of rural pastoral areas, is key to promoting the overall benefits of the cultural industry. Targeted investments and policy support for rural areas will be essential for optimizing Tibet's cultural industry development.

## Robustness and endogeneity tests

To ensure the robustness of the estimation results, this paper employs two testing methods (Refer to Table 11).

**Table 11. Robustness test analysis for replaced variables.**

| Variable | Replacement Variable |
|---|---|
| DIt | 3.2059*** |
| | 8.70 |
| Control Variables | Yes |
| Constant | 4.8184** |
| | 2.62 |
| Observations | 11 |
| R-squared | 0.929 |

*** p<0.01

** p<0.05

* p<0.1.

First, robustness was verified by substituting the dependent variables. In the study of cultural industries, scholars have used various output indicators, such as cultural industry output value [41], and the added value of the tertiary sector as key measures to assess the cultural industry's contribution to the regional economy [42]. Accordingly, this paper uses the added value of the tertiary sector as a substitute variable to provide a more comprehensive reflection of the cultural industry's overall output and economic impact.

Regarding asset dimensions, scholars often rely on fixed asset investment in the cultural industry to evaluate industry efficiency [43]. As such, this paper uses fixed asset investment in the culture, sports, and entertainment sectors as a substitute variable.

In addition, household expenditures on education, culture, and entertainment are commonly used to reflect how economic development drives cultural consumption, supported by previous research as a key economic variable [44,45]. This variable is therefore included in this study as a substitute indicator.

These variables are closely tied to the core performance of the cultural industry and provide a multi-dimensional assessment of its economic contribution. By logarithmically transforming these variables and applying equal proportional weighting, a new dependent variable was constructed. This approach ensures the economic contribution of the cultural industry is captured from various perspectives while addressing potential issues related to non-normal data distribution.

Second, this study accounts for the lagged effect of digital infrastructure construction by introducing a two-period lag of the core explanatory variable. The results indicate that the positive impact of digital infrastructure on the cultural industry remains significant at the 5% level, confirming the robustness of the model. (Refer to Table 12).

In addition, to diagnose and correct potential endogeneity bias in the regression model, this study employed the instrumental variable (IV) estimation method. The core explanatory variable may be subject to endogeneity due to its correlation with the error term. To address this issue, the study used one- and two-period lagged values of the core explanatory variable as instrumental variables and conducted an endogeneity test. The rationale for using these lagged values is that they are strongly correlated with the current value of digital infrastructure but uncorrelated with the error term, thereby satisfying the key assumptions of relevance and exogeneity required for IV estimation (Refer to Table 13).

To verify the validity of the selected instrumental variables, several tests were performed. First, the Cragg-Donald Wald F-statistic was 123.495, significantly exceeding the critical value of 19.93 for a 10% maximal IV relative bias, according to the Stock-Yogo test, indicating that

**Table 12. Robustness test analysis with a two-period lag.**

| Variable | Cultural Industry Productivity |
|---|---|
| Lagged DIt (Two Periods) | 1.1061** |
| | 2.79 |
| Control Variables | Yes |
| Constant | -1.7052 |
| | (-1.33) |
| Observations | 9 |
| R-squared | 0.894 |

*** $p < 0.01$

** $p < 0.05$, * $p < 0.1$.

**Table 13. Endogeneity test analysis of the impact of digital infrastructure on TFP in Tibet's cultural industry.**

| Variable | TFPOE |
|---|---|
| Lag 2-period DIt | 1.6813*** |
| Lag 1-period DIt | 0.5033** |
| Control Variables | Yes |
| Constant | -1.7915 (-0.59) |
| Observations | 10 |
| R-squared | 0.5114 |
| Cragg-Donald Wald F-statistic | 123.495 |
| Stock-Yogo Critical Value (10% Max IV Failure Rate) | 19.93 |
| Anderson Underidentification Test Chi-sq(1) P-value | 0.0019 |
| Sargan Test Chi-sq(1) P-value | 0.4232 |

*** $p < 0.01$

** $p < 0.05$, * $p < 0.1$.

the instrumental variables are not weak and are highly correlated with the core explanatory variable. Second, the p-value of the Anderson underidentification test was 0.0019, confirming that the model is properly identified. Finally, the Sargan test returned a p-value of 0.4232, supporting the null hypothesis that the instrumental variables are uncorrelated with the error term, thereby confirming their exogeneity. As a result, both the one-period and two-period lagged instrumental variables passed the tests for relevance and exogeneity, ensuring the reliability of the IV estimation results.

To further ensure the accuracy of the study's conclusions, the Sobel test was conducted to verify whether the proposed mediation mechanisms hold. The results, presented in the last row of Table 14, indicate that the Sobel test P-values for industrial agglomeration, logistics efficiency, and the urban-rural consumption gap were 0.013, 0.057, and 0, respectively. Each

**Table 14. Sobel test results.**

| Mediating Variable | Direct Effect | Indirect Effect | p-Value |
|---|---|---|---|
| Cultural Industry Agglomeration Degree | -0.369 | 2.229 | 0.013 |
| Goods Circulation Efficiency | 0.504 | 1.356 | 0.057 |
| Urban-Rural Consumption Gap | -2.652 | 4.512 | 0 |

passed the significance test at the 1% level, confirming that the mediating effects of industrial agglomeration, logistics efficiency, and the urban-rural consumption gap are statistically significant. (Refer to Table 14).

## Major conclusions, extended discussion, and policy recommendations

### Major conclusions

Drawing on data from Tibet's cultural industry between 2011 and 2021, this study empirically analyzed the impact of digital infrastructure on Total Factor Productivity (TFP). Several key conclusions were derived, validating the study's hypotheses. The findings are as follows:

Digital infrastructure significantly enhances the efficiency of the cultural industry: The study confirms Hypothesis 1, demonstrating that digital infrastructure facilitates the decentralization of the cultural industry, thereby improving overall efficiency. The results show that the development of digital infrastructure creates more opportunities for cultural products from remote areas to reach the market, optimizing information flow and resource allocation, which in turn substantially boosts productivity within the cultural sector [3,46].

Digital infrastructure promotes geographical decentralization of the cultural industry: Hypothesis 2 is validated, revealing that although digital infrastructure fosters geographical decentralization—contrary to traditional agglomeration theory—this decentralization enhances productivity in Tibet's unique environment. By improving communication and information flow, cultural enterprises in Tibet can successfully operate under decentralized conditions, thereby strengthening their competitiveness [1,4].

Digital infrastructure reduces the urban-rural cultural consumption gap: The study supports Hypothesis 3, showing that digital infrastructure improves network coverage and information flow in rural areas, narrowing the urban-rural consumption gap and fostering balanced cultural consumption. This result underscores that the construction of digital infrastructure effectively balances urban and rural cultural consumption, thereby improving the overall efficiency of the cultural industry [2,5].

### Extended discussion

Building on the findings of this study, we explore in greater depth the role of digital infrastructure in advancing Tibet's cultural industry. This section critically examines both the academic and practical implications of these mechanisms by comparing existing literature:

The influence of digital infrastructure on the efficiency of the cultural industry is particularly pronounced in Tibet's context. Gu Jiang and Chen Lu's research highlights how digital infrastructure optimizes information flow, thereby improving cultural industry efficiency [3]. However, this study extends their work by demonstrating that digital infrastructure not only enhances information flow but also plays a crucial role in narrowing the urban-rural consumption gap, fostering balanced regional cultural development.

The decentralized nature of digital infrastructure corresponds well with the unique characteristics of Tibet's cultural industry. Unlike traditional agglomeration theories, where geographic clusters are central to innovation, digital infrastructure supports a decentralized model that reduces dependency on physical proximity while promoting innovation and sustainable development [3]. This finding contrasts with the agglomeration effects discussed by Horkheimer et al., yet illustrates the adaptability of digital infrastructure in geographically distinct regions [46].

Logistics efficiency, while significant in other regions, has a more limited role in Tibet's cultural industry. Ji Yanlong's research suggests that digital infrastructure can significantly

enhance logistics efficiency [1]. However, this study reveals that Tibet's geographic challenges constrain the standardization of logistics. Therefore, tailored logistics solutions are essential for the region's cultural industry, underscoring the need for localized strategies in logistics optimization [5].

Digital infrastructure also helps reduce the urban-rural consumption gap. Cheng Chengping et al. demonstrate that digital technology can enhance cultural consumption in remote areas [2]. However, this paper further emphasizes that unequal investment in digital infrastructure could inadvertently widen the urban-rural economic divide. To mitigate this, future policies must prioritize equitable distribution of digital resources, particularly to support rural areas [6,7].

## Policy recommendations

Building on the theory of New-Quality Productivity, policy formulation should prioritize the innovative allocation of resources and production factors to amplify the impact of digital infrastructure within the cultural industry. Based on the study's findings, the following specific policy recommendations are proposed:

1. **Increase investment in digital infrastructure in remote areas:** To enhance the overall efficiency of Tibet's cultural industry, it is recommended that the government increase investment in digital infrastructure, particularly in rural and remote areas. This expansion should not be limited to urban centers but extend to less accessible regions, enabling the sharing of cultural resources between urban and rural areas and enhancing the overall reach of New-Quality Productivity. By boosting digital capabilities in rural regions, the urban-rural consumption gap can be further reduced, fostering more balanced development within the cultural industry [47,48].

2. **Develop a decentralized cultural industry chain:** In regions with complex geographical landscapes like Tibet, digital infrastructure policies should aim to cultivate a decentralized and flexible cultural industry chain. Encouraging small, personalized cultural enterprises to enter the market via digital platforms, alongside providing tailored technical and financial support, will stimulate local cultural industries. This approach aligns with the core principles of New-Quality Productivity by enhancing productivity through the innovative allocation of production factors [49].

3. **Enhance customization in logistics and supply chain management:** Although improving logistics efficiency may have limited impact in Tibet, policy should focus on developing customized logistics and supply chain management solutions tailored to the unique needs of the region's cultural industry. By offering personalized logistics support, especially for non-standardized cultural products, the efficiency of circulating these products in national and international markets can be improved, promoting the sustainable development of the cultural industry and driving New-Quality Productivity [50].

4. **Deepen digital empowerment and promote industry innovation:** Policies should strengthen the promotion and training related to digital technology applications, especially targeting the digital transformation needs of cultural enterprises. Emphasis should be placed on fostering deep integration between the cultural industry and digital technologies, enhancing data application capabilities, and driving innovation in digital services. This will not only enhance New-Quality Productivity but also ensure the sustainable development of the cultural industry in a globally competitive environment.

## Supporting information

**S1 Data.**
(XLSX)

## Author Contributions

**Conceptualization:** Yuanyuan Li.

**Data curation:** Yuanyuan Li.

**Formal analysis:** Yuanyuan Li.

**Funding acquisition:** Caidan Gazang.

**Investigation:** Qianqian Du.

**Methodology:** Yuanyuan Li.

**Resources:** Caidan Gazang.

**Software:** Yuanyuan Li.

**Supervision:** Caidan Gazang.

**Validation:** Yuanyuan Li.

**Visualization:** Yuanyuan Li.

**Writing – original draft:** Yuanyuan Li.

**Writing – review & editing:** Yuanyuan Li, Guihua Ma.

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
