## [Decision Letter · Decision Letter 0]

4 Sep 2024

PONE-D-24-29366Can Strengthening Digital Infrastructure Enhance Productivity in the Cultural Industry? Evidence from TibetPLOS ONE

Dear Dr. Li,

Thank you for submitting your manuscript to PLOS ONE. After careful consideration, we feel that it has merit but does not fully meet PLOS ONE’s publication criteria as it currently stands. Therefore, we invite you to submit a revised version of the manuscript that addresses the points raised during the review process.

We look forward to receiving your revised manuscript.

Kind regards,

Ying Wang

Academic Editor

PLOS ONE

**Journal requirements:**

This paper is a phase result of the "Practical Analysis of Tibet's Cultural Industry from a Digital Perspective," a project under the High-Level Talent Training Program for Graduate Students at Tibet University (Project No.: 2020-GSP-B037).

**Additional Editor Comments:**

Dear authors,

A review of your manuscript is now completed. The comments by editors and reviewers are listed at the end of this letter. I hope the reviewers' comments can help you to make your paper better to meet the publication requirement from the journal. Please revise your paper according to the comments by the reviewers.

When revising your manuscript, please consider EACH issue mentioned in the editors and reviewers' comments carefully: please outline every change made in response to their comments by highlighting the revisions with color text. You also need to submit a file of Response to Editor and Reviewers separately when you resubmit your paper.

Kind regards,

Ying Wang, Ph.D.

Academic Editor

PLOS ONE

Reviewers' comments:

Reviewer's Responses to Questions

**Comments to the Author**

1. Is the manuscript technically sound, and do the data support the conclusions?

Reviewer #1: Yes

Reviewer #2: Yes

2. Has the statistical analysis been performed appropriately and rigorously? 

Reviewer #1: Yes

Reviewer #2: No

3. Have the authors made all data underlying the findings in their manuscript fully available?

Reviewer #1: Yes

Reviewer #2: Yes

4. Is the manuscript presented in an intelligible fashion and written in standard English?

Reviewer #1: Yes

Reviewer #2: Yes

5. Review Comments to the Author

**Reviewer #1: **The author has insightfully explored the impact of the digital economy on the cultural industry, and the article is well-written and readable. However, there are areas that need further enhancement:

1.The concept of " New-Quality Productivity" is novel, whereas "total factor productivity" is efficiency-based. The author should clarify their applicability and inherent relationship in this article.

2.Section 2, titled "Theoretical Mechanisms and Research Hypotheses," lacks clear theoretical content. Are these new productive forces or total factor productivity? These are concepts or methods, not theories. It is suggested that the author revise this section's title and restructure the text's logic.

3.The formatting of the article needs consistency; for instance, the last three paragraphs of section 3.1.1 have shifted to left alignment.

4.The results are specific and rich, but the policy recommendations are vague. Recommendations should be based on the main findings to better highlight the article's theoretical and practical value.

5.The introduction and policy recommendations lack sufficient references.

**Reviewer #2:** This paper developed an empirical study to analyze how digital infrastructures can increase the productivity of cultural industries by adjusting industrial agglomeration, improving the efficiency of goods circulation and narrowing the consumption gap between urban and rural areas in Tibet, China. In general, this research fits in the main theme of the journal, but still requires several revisions. Below are some specific comments:

1. New quality productivity is a concept put forward by China's high-quality development, not a theory, so the author should correctly understand the expression.

2. The contribution of the article needs to be further highlighted.

3. The “Theoretical Mechanisms and Research Hypotheses” section needs to be strengthened with references to support the hypotheses.

4. The tables in the article are vague and are not placed in the appropriate place in the article.

5. Why does the sentence“Here is the English translation for your description of the use of statistical software and regression analysis:” appear in the article? Please critically check the content of the article expression

6. In table 8, what is the significance of A? Adding a note or applying the full name would be more beneficial to the reader's understanding

7. In table 9, the significance of * is not clearly indicated, does it mean a statistically significant level as in the previous table or is it a confidence level as described in your article?

8. In the Robust test, firstly, the authors substitute the dependent variable by replacing it, but are the two variables substitutable for each other? Is there any evidence for this? Secondly, the author applies IV to eliminate endogeneity, but does not verify the validity of IV, is the IV you chose appropriate?

9. The description of the results section of the article is not clear and concise enough, there are too many tables, and the formatting of the tables should be improved to make them clearer and more intuitive.

10. The article lacks further discussion of the main results, which need to be analyzed in the ‘Discussion’ section, considering literature comparisons.

11. The academic writing of this manuscript needs to be improved. There are a few problems including format, grammar, logic flow, specific writing style, etc. please check and revise accordingly.

6. PLOS authors have the option to publish the peer review history of their article (what does this mean?). If published, this will include your full peer review and any attached files.

Reviewer #1: No

Reviewer #2: No

---

## [Author Response · Author response to Decision Letter 0]

18 Oct 2024

Response to Editor and Reviewers

Dear Reviewers,

I would like to express my sincere gratitude to the reviewers for their insightful and constructive feedback. I highly value each of the comments provided and have carefully reviewed and addressed them one by one.

Reviewer 1,

1.Comment: The concept of "New-Quality Productivity" is novel, whereas "total factor productivity" is efficiency-based. The author should clarify their applicability and inherent relationship in this article.

Response: I accept this comment. I have clarified the relationship between "New-Quality Productivity" and TFP in the revised version. Specifically, I have revised the abstract to state that the study is based on the TFP framework, and corrected the description by emphasizing that "New-Quality Productivity" is a concept rather than a theory. Additionally, I have included further content explaining that the core aspect of New-Quality Productivity is the improvement of TFP. The introduction now includes six academic consensuses that clarify how TFP has evolved over time and has been adapted to China's context. I also explain that New-Quality Productivity is not an independent theoretical system but rather a key path to improving TFP through the innovative allocation of production factors. Furthermore, several new references have been added to the literature review to enhance this explanation.

2.Comment: Section 2, titled "Theoretical Mechanisms and Research Hypotheses," lacks clear theoretical content. Are these new productive forces or total factor productivity? These are concepts or methods, not theories. It is suggested that the author revise this section's title and restructure the text's logic.

Response: I have revised the title of Section 2 to "Theoretical Foundation and Mechanisms" and strengthened the theoretical content. Specifically, I added a discussion on TFP theory and common methods for evaluating it, with a focus on how these methods are applied to the analysis of the Tibetan cultural industry. This has strengthened the theoretical grounding and improved the overall logical structure of this section.

3.Comment: The formatting of the article needs consistency; for instance, the last three paragraphs of section 3.1.1 have shifted to left alignment.

Response: I have corrected the formatting inconsistencies throughout the manuscript, ensuring that all sections are properly aligned and formatted consistently.

4. Comment: The results are specific and rich, but the policy recommendations are vague. Recommendations should be based on the main findings to better highlight the article's theoretical and practical value.

Response: I have revised the policy recommendations to align more closely with the main findings. Specifically, the recommendations now directly address the issues identified in the analysis and are clearer and more precise. For instance, I added suggestions for enhancing digital infrastructure investment in rural and remote areas, based on the analysis of cultural product accessibility. Additionally, I emphasized the development of small-scale, personalized enterprises in regions like Tibet, where cultural businesses are geographically dispersed. These revisions ensure that the policy suggestions are tightly connected to the theoretical framework of New-Quality Productivity.

4.Comment: The introduction and policy recommendations lack sufficient references.

Response: I have added 7 new references to the introduction and 4 references to the policy recommendations to substantiate my conclusions and to align the policy suggestions with existing expert perspectives.

Reviewer 2,

1.Comment: New quality productivity is a concept put forward by China's high-quality development, not a theory, so the author should correctly understand the expression.

Response: I have revised the abstract to clarify that New-Quality Productivity is a concept rather than a theory, and added a more detailed explanation in the introduction. The introduction now explicitly discusses how New-Quality Productivity is tied to TFP improvement and how it fits into China’s broader development strategy, particularly through digital infrastructure development.

2.Comment: The contribution of the article needs to be further highlighted.

Response: I have expanded the introduction to better highlight the article’s contributions, particularly focusing on how digital infrastructure optimizes the allocation of production factors and improves TFP. I also emphasized that while many previous studies have stressed the importance of infrastructure in Tibet, few have explored the specific mechanisms of its impact on various aspects of the cultural industries, which this paper addresses.

3.Comment: The “Theoretical Mechanisms and Research Hypotheses” section needs to be strengthened with references to support the hypotheses.

Response: I have revised this section to "Theoretical Foundation, Mechanism Analysis, and Research Hypotheses" and added 5 new references to support the hypotheses, thereby strengthening the theoretical foundation of the analysis and enhancing its academic rigor.

4.Comment: The tables in the article are vague and are not placed in the appropriate place in the article.

Response: I have reformatted all tables using three-line tables for consistency and clarity, and removed unnecessary content or calculations. Each table has been placed in the appropriate section to align with the corresponding analysis for better flow and reader comprehension.

5.Comment: Why does the sentence "Here is the English translation for your description of the use of statistical software and regression analysis:" appear in the article? Please critically check the content of the article expression.

Response: I have reviewed the entire article and removed this unintended sentence. The entire manuscript has been carefully checked for content and language errors.

6.Comment: In Table 8, what is the significance of A? Adding a note or applying the full name would be more beneficial to the reader's understanding.

Response: I have reviewed all tables, including Table 8, and ensured that all abbreviations, such as "A," have been replaced with their full names or clearly explained in footnotes to enhance clarity for the readers.

7.Comment: In Table 9, the significance of * is not clearly indicated. Does it mean a statistically significant level as in the previous table, or is it a confidence level as described in your article?

Response: I have added the missing note explaining that the * in Table 9 indicates statistical significance, consistent with the usage in the previous table and the rest of the manuscript.

8.Comment: In the Robust test, the authors substitute the dependent variable by replacing it, but are the two variables substitutable for each other? Is there any evidence for this? Secondly, the author applies IV to eliminate endogeneity, but does not verify the validity of IV. Is the IV you chose appropriate?

Response: Regarding the substitution of the dependent variable, I have added 5 empirical papers on cultural industries as supporting evidence, demonstrating that dimensions such as cultural industry output and fixed asset investment are valid alternative measures. For endogeneity, I introduced a new instrumental variable (IV) using the lagged value of digital infrastructure. I verified its validity using Cragg-Donald Wald F statistics, Anderson underidentification tests, and Sargan tests. The F-statistic (123.495) exceeds the critical value, indicating that the IV is not weak. The Anderson underidentification test has a P-value of 0.0019, showing the model is properly identified. The Sargan test's P-value of 0.4232 confirms the exogeneity of the IV.

9.Comment: The description of the results section of the article is not clear and concise enough, there are too many tables, and the formatting of the tables should be improved to make them clearer and more intuitive.

Response: I have simplified the description in the results section, avoiding excessive explanation and improving the logical flow. The tables have been reformatted using three-line tables for clarity, and unnecessary content has been removed. All tables have been repositioned to align with the text for better readability.

10.Comment: The article lacks further discussion of the main results, which need to be analyzed in the "Discussion" section, considering literature comparisons.

Response: I have added a new section titled “4.2 Extended Discussion,” where I compare my findings with 8 relevant studies to provide a broader context. This includes discussing the impact of digital infrastructure on the urban-rural consumption gap, the opposition of dispersed cultural enterprises to agglomeration theory, the prioritization of customized logistics needs over mere logistics efficiency, and the imbalance of investment between urban and rural areas.

11.Comment: The academic writing of this manuscript needs to be improved. There are a few problems including format, grammar, logic flow, and specific writing style. Please check and revise accordingly.

Response: I have thoroughly reviewed the manuscript for grammatical issues, logic flow, and writing style. I have made the necessary revisions to improve the clarity and academic tone of the manuscript. However, I acknowledge that there may still be areas for improvement, and I welcome further feedback from the reviewers and editors.

Above all, these suggestions have greatly contributed to enhancing the quality of my work, and I have learned a lot through this process. I hope that the revisions I have made will meet the high standards of the journal and align with the expectations of both the reviewers and the editor. Thank you for your consideration of the revised manuscript.

Sincerely,

Li Yuanyuan, Du Qianqian, Gazang Caidan

---

## [Decision Letter · Decision Letter 1]

12 Nov 2024

PONE-D-24-29366R1Can Strengthening Digital Infrastructure Enhance Productivity in the Cultural Industry? Evidence from TibetPLOS ONE

Dear Dr. Li,

Thank you for submitting your manuscript to PLOS ONE. After careful consideration, we feel that it has merit but does not fully meet PLOS ONE’s publication criteria as it currently stands. Therefore, we invite you to submit a revised version of the manuscript that addresses the points raised during the review process. Please revise your manuscript based on the comments from reviewers and submit your revised manuscript by Dec 27 2024 11:59PM. If you will need more time than this to complete your revisions, please reply to this message or contact the journal office at plosone@plos.org. Please include the following items when submitting your revised manuscript:A rebuttal letter that responds to each point raised by the academic editor and reviewer(s). You should upload this letter as a separate file labeled 'Response to Reviewers'.A marked-up copy of your manuscript that highlights changes made to the original version. You should upload this as a separate file labeled 'Revised Manuscript with Track Changes'.An unmarked version of your revised paper without tracked changes. You should upload this as a separate file labeled 'Manuscript'.If applicable, we recommend that you deposit your laboratory protocols in protocols.io to enhance the reproducibility of your results. Protocols.io assigns your protocol its own identifier (DOI) so that it can be cited independently in the future. For instructions see: https://journals.plos.org/plosone/s/submission-guidelines#loc-laboratory-protocols. Additionally, PLOS ONE offers an option for publishing peer-reviewed Lab Protocol articles, which describe protocols hosted on protocols.io. Read more information on sharing protocols at https://plos.org/protocols?utm_medium=editorial-email&utm_source=authorletters&utm_campaign=protocols.

We look forward to receiving your revised manuscript.

Kind regards,

Ying Wang

Academic Editor

PLOS ONE

Journal Requirements:

Reviewers' comments:

Reviewer's Responses to Questions

**Comments to the Author**

1. If the authors have adequately addressed your comments raised in a previous round of review and you feel that this manuscript is now acceptable for publication, you may indicate that here to bypass the “Comments to the Author” section, enter your conflict of interest statement in the “Confidential to Editor” section, and submit your "Accept" recommendation.

Reviewer #1: (No Response)

Reviewer #2: All comments have been addressed

2. Is the manuscript technically sound, and do the data support the conclusions?

Reviewer #1: Yes

Reviewer #2: Yes

3. Has the statistical analysis been performed appropriately and rigorously? 

Reviewer #1: Yes

Reviewer #2: Yes

4. Have the authors made all data underlying the findings in their manuscript fully available?

Reviewer #1: Yes

Reviewer #2: Yes

5. Is the manuscript presented in an intelligible fashion and written in standard English?

Reviewer #1: Yes

Reviewer #2: Yes

6. Review Comments to the Author

Reviewer #1: The author has made significant efforts to improve this article, but it seems that there is still some confusion between the concepts of theory and methodology. Total Factor Productivity (TFP) is not a theoretical framework, but rather a concept used in economics to measure the efficiency of all inputs in the production process. TFP is based on theoretical foundations, such as growth theory.

Reviewer #2: (No Response)

7. PLOS authors have the option to publish the peer review history of their article (what does this mean?). If published, this will include your full peer review and any attached files.

Reviewer #1: No

Reviewer #2: No

---

## [Author Response · Author response to Decision Letter 1]

5 Dec 2024

Dear reviewers,

Thank you for the thoughtful feedback. We sincerely accept the suggestions and have made the necessary revisions in the revised manuscript. Specifically, we have:

Added two additional theoretical foundations to enhance the study's theoretical framework.

Clarified the conceptual positioning of TFP as a measurement tool for production efficiency, ensuring it is distinguished from theoretical concepts to avoid confusion.

Explicitly differentiated theoretical frameworks from methodological tools to ensure conceptual clarity.

These updates are detailed in the Response to Editor and Reviewer document and highlighted in the Revised Manuscript with Track Changes (red text).

We appreciate your valuable feedback and hope that the revisions meet your expectations.

Sincerely,

Yuanyuan Li

---

## [Decision Letter · Decision Letter 2]

27 Dec 2024

Can Strengthening Digital Infrastructure Enhance Productivity in the Cultural Industry? Evidence from Tibet

PONE-D-24-29366R2

Dear Dr. Yuanyuan Li,

We’re pleased to inform you that your manuscript has been judged scientifically suitable for publication and will be formally accepted for publication once it meets all outstanding technical requirements.

Kind regards,

Ying Wang

Academic Editor

PLOS ONE

Additional Editor Comments (optional):

Reviewers' comments:

Reviewer's Responses to Questions

**Comments to the Author**

1. If the authors have adequately addressed your comments raised in a previous round of review and you feel that this manuscript is now acceptable for publication, you may indicate that here to bypass the “Comments to the Author” section, enter your conflict of interest statement in the “Confidential to Editor” section, and submit your "Accept" recommendation.

Reviewer #1: All comments have been addressed

Reviewer #2: All comments have been addressed

2. Is the manuscript technically sound, and do the data support the conclusions?

Reviewer #1: Yes

Reviewer #2: Yes

3. Has the statistical analysis been performed appropriately and rigorously? 

Reviewer #1: Yes

Reviewer #2: Yes

4. Have the authors made all data underlying the findings in their manuscript fully available?

Reviewer #1: Yes

Reviewer #2: (No Response)

5. Is the manuscript presented in an intelligible fashion and written in standard English?

Reviewer #1: Yes

Reviewer #2: (No Response)

6. Review Comments to the Author

Reviewer #1: The author's revisions are thorough and comprehensive, with significant scientific value. Publication is recommended.

Reviewer #2: (No Response)

7. PLOS authors have the option to publish the peer review history of their article (what does this mean?). If published, this will include your full peer review and any attached files.

Reviewer #1: No

Reviewer #2: No

---

## [Editor Report · Acceptance letter]

13 Jan 2025

PONE-D-24-29366R2 

PLOS ONE

Dear Dr. Li, 

I'm pleased to inform you that your manuscript has been deemed suitable for publication in PLOS ONE. Congratulations! Your manuscript is now being handed over to our production team.

Kind regards, 

on behalf of

Dr. Ying Wang 

Academic Editor

PLOS ONE